# Effects of Leptin, Growth Hormone and Photoperiod on Pituitary SOCS-3 Expression in Sheep

**DOI:** 10.3390/ani12030403

**Published:** 2022-02-08

**Authors:** Dorota Anna Zieba, Malgorzata Szczesna, Katarzyna Kirsz, Weronika Biernat

**Affiliations:** Department of Animal Nutrition and Biotechnology, and Fisheries, Faculty of Animal Sciences, University of Agriculture in Krakow, 31-120 Krakow, Poland; malgorzata.szczesna@urk.edu.pl (M.S.); katarzyna.kirsz@urk.edu.pl (K.K.); weronika.biernat@urk.edu.pl (W.B.)

**Keywords:** leptin, leptin resistance, GH, SOCS-3, photoperiod, sheep

## Abstract

**Simple Summary:**

Maintaining energy homeostasis requires numerous processes and interactions between many systems. This study investigated the relationships between leptin, growth hormone (GH), and factors involved in cell signaling, such as suppressors of cytokine signaling-3 (SOCS-3). Exogenous or endogenous factors affecting these relationships may have different origins, and their interactions are determined in this study. The finding that the length of day significantly influenced the concentrations of growth hormone as well as pituitary leptin- and GH-dependent SOCS-3 expressions suggests that photoperiod plays an important role in regulating the physiological processes underlying adaptive phenomena in response to changing seasons in sheep, which facilitates energy homeostasis despite changing external and internal conditions.

**Abstract:**

This study examined how leptin affects growth hormone (GH) release and investigated the effects of leptin, GH, and day length on the suppressor of cytokine signaling-3 (SOCS-3) mRNA levels in the adenohypophyses of sheep. The study consisted of two experiments. The first experiment was conducted during long (LD) and short (SD) days. Within-season and replicate sheep were centrally infused with Ringer-Locke buffer or leptin three times at 60-min intervals at the beginning of experiments. The second experiment involved adenohypophyses collected from sheep that were euthanized in May or November. Pituitary explants were treated with medium alone (Control) or medium with leptin or GH at different concentrations and incubated for various times. The results of the first experiment indicated GH concentrations were seasonally dependent and that leptin had no effect on GH secretion. The results of the second experiment indicated a stronger influence of leptin on the expression of SOCS-3 during the SD season than the LD season. During SDs, significant effects of both GH doses on SOCS-3 expression were observed. These results indicate a strong association between leptin, GH, and SOCS-3, which may explain the disruption of SOCS-3 leptin and GH signaling and the dominant effect of photoperiod on the above relationships.

## 1. Introduction

Leptin is produced by many cells and in highest amounts by white adipose tissue cells. This hormone acts on hunger and satiety centers in the hypothalamus, where its main action is to suppress appetite, and it is an important regulator of numerous physiological processes, including reproduction and metabolism.

Growth hormone (GH) stimulates cell growth, division, and regeneration and is, thus, a crucial regulator of cell metabolism that strongly influences many processes within organisms. In mammals that demonstrate evidence of seasonality (such as sheep), GH secretion is regulated by photoperiod [1].

Leptin may play a role in the regulation of GH secretion; however, the existing information is ambiguous. Iqbal et al. [2] demonstrated the presence of leptin receptors (LRs) on ovine somatotrophs. In rats, an intracerebroventricular (icv) infusion of leptin demonstrated that this hormone decreased the expression of somatostatin and increased the expression levels of growth hormone releasing hormone (GHRH), which corresponded to an increase in the levels of GH mRNA [3]. This result suggests that leptin can directly stimulate GH secretion from pituitary explants [4,5,6].

In sheep, the effect of leptin on GH secretion in vivo and in vitro may also depend on nutritional status, as shown in previous experiments [4,7,8,9].

The intracellular signaling pathways of leptin and GH can be positively and negatively regulated. One of the major negative regulators of these pathways is the suppressor of cytokine signaling-3 (SOCS-3) protein, which can suppress the actions of both GH [10,11,12] and leptin [13]. Transcription of the SOCS-3 gene is not constitutive in a majority of tissues (except in the brain), and thus specific factors are needed to induce its transcription. SOCS-3 and many factors that induce its expression are localized in the arcuate nucleus (ARC), paraventricular nucleus (PVN), ventromedial and dorsomedial nuclei, suprachiasmatic nucleus [11,14], and the pituitary gland, suggesting that SOCS-3 may play a pivotal role in modulating neuroendocrine interactions among hormones, such as leptin, prolactin (PRL), and resistin [12]. Data suggest that SOCS-3 plays a significant role in the pituitary gland. Research in recent years has shown that expression of SOCS-3 also depends on environmental stimuli, such as photoperiod and food availability [14].

In mice lacking the functional gene encoding leptin, a high level of leptin-induced SOCS-3 expression has been reported in the PVN of the hypothalamus; this expression maybe involved in leptin resistance or leptin insensitivity [14,15]. Experiments in mammals that demonstrate evidence of seasonality have shown that the abundance of SOCS-3 transcripts and SOCS-3 protein levels are dependent on environmental influences, including the day length [14,15]. Tups et al. [14] found that SOCS-3 expression is higher during long days (LDs) and reduced during short days (SDs) in the Siberian hamster (*Phodopus sungorus*). Additionally, the leptin-induced stimulation of SOCS-3 expression was reduced during SDs [14,16]. An increase in SOCS-3 expression in response to leptin may be a key cause of resistance/insensitivity to the anorectic action of this hormone, a phenomenon that is pathological in obese individuals and typical in sheep during seasons with LDs. The seasonality of leptin resistance in sheep is associated with the adaptation of these animals to annual changes in the demand and supply of energy, but the neuroendocrine basis of this phenomenon is still not fully understood. Our previous studies [11,17,18] indicated a role of SOCS-3 in modulating the sensitivity of the hypothalamus and pituitary to leptin in vivo, according to the season. Although these data suggested that ICV infusions of leptin may alter the expression levels of SOCS-3 in sheep, they did not explain how the hormone exerts this action.

Previous studies on cattle have shown that leptin acts directly on the anterior pituitary to modulate GH release and that this effect was dependent upon the individual’s nutritional history [19]. Leptin modulated GH secretion in anterior pituitary explants collected from normal-fed and fasted cows. However, the effects of leptin in explants collected from normal-fed animals were seen only during GHRH stimulation [19].

Given the conflicting data on the effect of leptin on GH secretion in ruminants and considering sheep seasonality, the first goal of the present study was to examine how central treatment with leptin affects GH concentration in normally fed sheep under different photoperiodic conditions.

The second goal was to determine the in vitro influence of leptin and GH on SOCS-3 mRNA levels in adenohypophysis explants and to investigate whether the effects of these hormones were season-specific.

## 2. Materials and Methods

The 2nd Institutional Animal Care and Use Committee (IACUC) in Krakow evaluated and approved all experimental procedures (protocol no. 108/2018).

### 2.1. Animal Characteristics

The Polish Longwool ewe, a breed that exhibits strong seasonality in reproduction [20], was used in these studies. Polish Longwool ewes enter the breeding season when days shorten. The first estrous cycles usually occur in early August, and the breeding season ends in March. Traditionally, in Poland, these sheep are bred from September to October. The anestrous season lasts from March to August. Animals used in this study were 2 to 3 years old and had a mean body weight of 60 ± 5 kg. All animals were housed in individual pens under natural photoperiodic and thermoperiodic conditions (longitude: 19°57′ E, latitude: 50°04′ N) and had a moderate body condition score (BCS) of 3 (on a scale from 0 to 5; where 0 = emaciated and 5 = obese; [21]). The sheep were fed twice daily at 07:00 and 16:00 h with a diet formulated to supply 100% of the nutritional requirements for maintenance, according to the National Research Institute of Animal Production Recommendations [22]. Water was available ad libitum.

### 2.2. Experiment I—The In Vivo Influence of Central Leptin Treatment on GH Concentrationin Sheep

#### 2.2.1. Animals and Treatments

Twenty-four ewes were prepared for experiments (ovariectomized and given estrogen replacement) as described in detail by Biernat et al. [23] to prevent cyclic alterations in the secretion of gonadal steroids that could create differences between females. Four weeks after ovariectomy, ewes were fitted surgically with third ventricle (IIIV) cannulas, according to the methodology reported by Traczyk and Przekop [24]. The location and functionality of the cannulas were verified by the continuous flow of cerebrospinal fluid (CSF). The sheep were given at least 3 weeks to recover from neurosurgery. Before the study, the ewes were familiarized with the experimental conditions by frequently placing them in individual wooden carts as described previously [23]. The experiments were performed in two different seasons: during longer days (spring: March, April, and May) and shorter days (autumn: September, October, and November), with 12 randomly selected ewes used in both seasons. In the morning of the day of each experiment, sheep were fitted with jugular vein catheters (central and peripheral venous catheters, Careflow^TM^, Argon, Billmed Sp. z o.o., Warsaw, Poland) for intensive blood sampling to determine the GH profile.

Before the experiments, polyethylene tubing (70 mm; 0.58 i.d., 0.96 mm o.d.; Intramedic Clay Adams Brand, Becton Dickinson, Sparks, MD, USA) was inserted using an aseptic technique through each IIIV guide cannula, with the distal end projected 3–5 mm past the end of the cannula and into the ventricle. The proximal end of the tubing extended 5–10 mm above the tip of the guide cannula. The tubing was adjusted until CSF flowed easily using a blunt 22 G needle and tuberculin syringe. The tubing was then plugged until later use. On the afternoon of each experimental day, the ewes were assigned randomly to one of three groups (*n* = 4 per group) and placed into carts, and CSF flow was confirmed.

#### 2.2.2. Treatments

Recombinant ovine leptin (roleptin) was purchased from Ray Biotech, Inc. (Norcross, GA, USA). The experiments were carried out on three groups of sheep:(1) the Control group, treated with Ringer-Locke buffer (pH 7.4); (2) the Leptin 1 group, treated with 0.5 μg/kg BW roleptin; and (3) the Leptin 2 group, treated with 1.0 μg/kg BW roleptin. The study was carried out in a switchback design such that all ewes (*n* = 4) in each group were given one of the subsequent infusions (Control, Leptin 1, or Leptin 2) directly into the IIIV in a random order, with intervals of approximately 14 days. Allocating 12 sheep to each of the two photoperiods (experiments performed in spring or autumn) yielded three replicates for each group. The dose of roleptin for this experiment was based on numerous previous experiments by our group [19,25,26] and information from the literature on studies with IIIV-cannulated sheep [27]. In the experiments, roleptin was injected at three time points: 0, 60, and 120 min after the experiment began. Blood samples (5 mL) were collected every 15 min for 6 h, beginning immediately before the first infusion (Figure 1). The blood samples were dispensed into tubes containing 150 mL of a solution containing heparin (10,000 IU/mL) and 5% (*w*/*v*) EDTA and were immediately placed on ice. Plasma was separated by centrifugation and stored at −20 °C until GH analysis was performed.

#### 2.2.3. RIA for GH

A radioimmunological assay was conducted by the double antibody method using anti-bovine GH and anti-rabbit-γ-globulin antisera, according to Dvorak et al., to assess plasma concentrations of GH [28]. Bovine GH (NIDDK-GH-B-1003A) served as a reference standard. The assay sensitivity was 0.4 ng/mL, and the intra- and interassay coefficients of variation were 5.45% and 10.20%, respectively.

#### 2.2.4. Data Analysis and Statistical Tests

The hormone concentrations were analyzed using the general linear models (GLM) procedure (PROC GLM). For the hormone comparisons, the overall ANOVA included the treatment (Control, Leptin 1, or Leptin 2), season, replicate within the season, the time within the season, and all two- and three-way interactions for the repeated measures in the switchback design. Significant treatment × season interactions resulted in a within-season model that included the treatment, time point, and treatment × time point. If the *F* test was significant, the Pdiff procedure of SAS was used to contrast the mean values. If the *F* value was significant, the means were contrasted using Duncan’s multiple-range test. Differences with *p* < 0.05 were considered statistically significant. The data are expressed as the means ± S.E.M.

### 2.3. Experiment II—The In Vitro Influence of Leptin and GH on SOCS-3 Transcript Levels in Adenohypophysis Explants

#### 2.3.1. Animals and Treatments

In this experiment eight sheep were used. The ewes were euthanized humanely, and the tissues were aseptically isolated from the ewes 10–15 min post-mortem.

Brains with an intact infundibulum were quickly removed from the skulls of all sheep and frozen on dry ice. The pituitaries were harvested from the sella turcica. Samples of the adenohypophyses were aseptically obtained from four ewes per experimental season; four ewes were randomly selected during spring (May—longdays), and an additional four ewes were randomly selected during the autumn (November—shortdays). The pituitaries were transported to the laboratory, and all subsequent procedures were performed under sterile conditions. Before incubation, the pituitaries were rinsed three times in Parker’s medium (M199; Laboratory of Vaccines, Lublin, Poland). The tissues were cut into approximately 50-mg explants. Incubations were performed in a 6-well Corning tissue culture dish (Corning Glass Works, New York, NY, USA). The explants were located on a gas–liquid interface as described in detail above [29] and kept in M199 (a total volume of 2.5 mL per well). Explants were incubated in an atmosphere with 5% CO_2_ and 95% humidified air at 37 °C for a maximal time of 2 h. The pituitary explants were treated with medium alone (control), medium enriched with GH (100 ng/mL or 300 ng/mL; Sigma–Aldrich Corporation, St. Louis, MO, USA), or medium enriched with roleptin (50 ng/mL or 100 ng/mL; Ray Biotech Inc., Norcross, GA, USA). Each treatment group had three replicates. Every 60 min, one milliliter of medium was collected from the treatment group and the same volume of fresh medium was added. The adenohypophysis explants were incubated for four different intervals: 0, 30, 60, and 120 min. At the end of each interval, parts of the pituitary gland were placed in phosphate buffered saline (PBS; Vaccine Laboratory, Lublin, Poland), dried on sterile tissue paper, frozen in liquid nitrogen, and kept at −80 °C for further analyses.

#### 2.3.2. Molecular Analysis

To measure SOCS-3 mRNA levels, real-time PCR was performed. The detailed procedure is described in a paper by Zieba et al. [11]. Primers and probes were designed using Primer Express software v. 2.0 (Applied Biosystems; Foster City, CA, USA) and are characterized in Table 1. Data were recorded with Applied Biosystem 7300 Real-Time PCR System SDS software.

#### 2.3.3. Molecular Data Analysis

Expression levels were calculated by relative quantification (RQ) analysis. In brief, the amplification plot provided by this analysis is a plot of fluorescence versus PCR cycle number. The Ct value is the fractional PCR cycle number where the fluorescent signal reaches the detection threshold. Therefore, the input cDNA copy number and Ct were inversely related. Data were analyzed using the 2^−ΔΔCt^ method, and Ct values were converted to fold-change RQ values. The RQ values from each gene were used to compare target gene expression across all groups. The mean mRNA expression levels of SOCS-3 in each sample were standardized against the expression of a reference gene (cyclophilin; CPH) and expressed relative to the calibrator sample. The variation in the Ct values for CPH among the treatment groups was not significant (*p* > 0.05). The mean ΔCt value for tissue collected from the control group was used to calibrate changes in the target gene expression among all treatment groups in the indicated season.

Differences in the means were compared with SigmaPlot statistical software (version 11.0; Systat Software Inc., Richmond, CA, USA) using pairwise multiple comparison procedures (Tukey test), preceded by the determination of a significant *F* value. Differences were considered statistically significant when *p* < 0.05.

## 3. Results

### 3.1. Experiment I—The In Vivo Influence of Central Leptin Treatment on GH Concentrations in Sheep

Depending on the day-length, plasma GH concentration was enhanced in the control group (*p* < 0.01) in the LD season compared with the SD season (Figure 2). There was no effect of leptin on GH concentrations during both LD and SD seasons (*p* > 0.5).

### 3.2. Experiment II—The In Vitro Influence of Leptin and GH on SOCS-3 Transcript Levels in Adenohypophysis Explants

Growth hormone and leptin at the previously mentioned concentrations affected the abundance of SOCS-3 transcripts in pituitary explants. The dose, incubation time, and season in which the pituitary explants were obtained produced notable effects. There were no differences (*p* > 0.1) in basal SOCS-3 expression levels in nontreated pituitaries that were frozen immediately after tissue collection during LDs and SDs.

#### 3.2.1. Leptin Effects

The leptin treatment results are shown in Figure 3. Low doses of leptin significantly (*p* < 0.01) increased the abundance of SOCS-3 transcripts during the LD season, but during the SD season, a significant (*p* < 0.01) effect of leptin was found only in explants incubated with the higher leptin dose. The effect of leptin was related to the length of incubation. In the LD season, (Figure 4a) a low dose of leptin significantly increased the abundance of SOCS-3 transcripts in pituitary gland explants incubated for 30 min (*p* < 0.001) and 120 min (*p* < 0.01), but no difference was found in explants incubated for 60 min. In addition, there were significant differences (*p* < 0.05) in SOCS-3 expression levels between the two leptin concentration groups after 120 min of incubation.

However, in the SD season (Figure 4b), leptin at a higher dose (100 ng/mL) significantly increased the expression of SOCS-3, which was especially true for samples incubated for 120 min. Roleptin at 100 ng/mL, compared to 50 ng/mL, affected the abundance of SOCS-3 transcripts more strongly (*p* < 0.001). Incubation of explants for 120 min resulted in a nearly 3-fold (Leptin 1, *p* < 0.001) and approximately 4-fold (Leptin 2, *p* < 0.001) rise in the abundance of SOCS-3 transcripts compared with that of the control. The level of SOCS-3 transcripts was higher than that of the control when the low dose of leptin was used, whereas the abundance of SOCS-3 transcripts significantly decreased after leptin treatment at 100 ng/mL (*p* < 0.01) and 60 min of incubation. Interestingly, a lower dose of leptin in the 30-min incubation significantly lowered the expression level of the examined factor by nearly 10-fold compared to that of the control (*p* < 0.001).

#### 3.2.2. Growth Hormone Effects

Under LD conditions, both doses of GH significantly (*p* < 0.001) inhibited SOCS-3 expression (Figure 5). In the SD condition, a dose-dependent effect of GH was noted. GH at the lower dose (100 ng/mL) significantly stimulated SOCS-3 expression (*p* < 0.05), whereas the higher dose (300 ng/mL) markedly (*p* < 0.001) increased the SOCS-3 transcript level compared with those of the control and a lower dose of GH (*p* < 0.05) (Figure 5).

In tissue collected under LD conditions (Figure 6a), GH strongly (*p* < 0.001) inhibited the abundance of SOCS-3 transcripts (after 30 and 60 min of incubation, and at 100 and 300 mg/mL doses). Moreover, significant (*p* < 0.01) differences between the two GH concentrations were noted after the incubation of the explants for 60 and 120 min. A concentration of 300 ng/mL GH decreased the abundance of SOCS-3 transcripts.

For tissue collected in the SD season (Figure 6b), the lower dose of GH after 30 min (*p* < 0.01) and 60 min (*p* < 0.01) of incubation lowered SOCS-3 transcript levels, but after 120 min, this dose significantly (*p* < 0.001) increased its level. The higher dose (300 ng/mL) of GH increased SOCS-3 transcript levels, and this increase was characterized in a time-dependent manner. After 60 min of incubation, there was an increase of approximately 30% in SOCS-3 mRNA levels (*p* < 0.05), and after 120 min of incubation there was nearly a 5-fold enhancement of SOCS-3 mRNA (*p* < 0.001). The SOCS-3 mRNA levels were higher in explants incubated with GH at a concentration of 300 ng/mL than in explants incubated with GH at a concentration of 100 ng/mL for every time point, and these differences were significant (*p* < 0.01).

## 4. Discussion

The results of these experiments indicated a predominant influence of photoperiod on the concentration of GH in sheep fed ad libitum and on the investigated interactions, under in vitro conditions. The first experiments showed no significant effect of leptin on GH concentrations, both during the LD and SD seasons. However, during the LD season, GH concentrations were significantly higher than those during the SD season. In sheep, the effect of leptin on GH release has previously been reported by Henry et al. [8], both in individuals undergoing dietary restrictions and in normally fed sheep. However, in their previous studies, the authors found no effect of leptin infusions on GH secretion in normally fed animals [7]. Interestingly, the effect of leptin established by Henry et al. [7] was consistent with those observed in the present paper. Our previous study demonstrated contradictory results on the effects of leptin on anterior pituitary explants in normally fed and fasted cows. Pituitary explants from normal-fed cows exhibited reduced GH secretion with most of the leptin doses when compared with control tissues [19] and explants collected from fasted cattle. The observed increase in GH secretion from the pituitary explants from fasted cows and the decreased secretion of GH from the tissues of normal-fed cows observed in that study could have resulted from differences in leptin receptor expression and a down-regulation of its expression in explants from well-fed cows or disturbance of the signaling pathway caused by higher expression of SOCS-3 in somatotrophs.

SOCS-3 has been suggested to play a primary role in modulating the interactions among proteins of the cytokine superfamily. SOCS protein expression is typically low; however, it can be influenced by many external and internal factors, including photoperiod and other hormones (i.e., PRL, resistin) [14,30]. Morrison et al. [9] observed that photoperiod influenced the sensitivity of the pituitary to leptin. Cytokines may reduce the levels of SOCS mRNA transcription. This reduction was demonstrated by Bjørbæk et al. [13,15], who first discovered the role of SOCS-3 in the leptin signaling pathway. Many scientists consider SOCS-3 to be a factor that inhibits the leptin-induced signal transduction and that could potentially mediate leptin insensitivity [31,32].

Tups et al. [14], in studies on Siberian hamsters, demonstrated that leptin increased SOCS-3 mRNA expression in the ARC, without determining SOCS-3 expression in the pituitary gland. The previous results of our group demonstrated contrasting effects of leptin on the expression levels of SOCS-3 in the ovine hypothalamus and pituitary in animals centrally infused with leptin [17,33] compared with the effects found in the present experiments. Furthermore, other experiments have noted that the relationship between leptin and SOCS-3 expression is dependent on the length of day. Although differences in SOCS-3 expression after a central infusion of leptin were noted at the level of the ovine pituitary gland and hypothalamus, leptin modulatedSOCS-3 mRNA levels in the pituitary only during the SD season [18]. In contrast, its influence in the hypothalamus was observed exclusively during the LD season [17].

The results of the present study demonstrated that the effect of leptin on the expression of SOCS-3 is stronger during the SD season. During this season, leptin at a dose of 100 ng/mL resulted in a two-fold increase in the SOCS-3 expression level, whereas leptin at a dose of 50 ng/mL only increased it by 38% (this increase was not statistically significant). The lack of evidence for a stimulating effect of the lower dose of leptin was likely the result of a lower expression of SOCS-3 in the SD season noted at the beginning of study.

The influence of leptin on SOCS-3 mRNA levels may depend on the species and tissues tested, the metabolic state of the organism, and the season in which the experiment was conducted. The results obtained in the present experiment are consistent with the results of a previous study and the present consensus of scientific investigations. We found a direct stimulating effect of leptin on SOCS-3 transcript levels in the adenohypophyses of sheep. Furthermore, leptin had more potent effects during the SD season than the LD season. We also demonstrated that the effect of leptin is evident as early as 30 min after treatment. The results demonstrate a distinct interaction between leptin and the leptin signaling inhibitor SOCS-3, which suggests that SOCS-3 created leptin insensitivity.

Depending on the tissue tested, GH can induce the expression of different proteins from the SOCS family [33].GH receptors, e.g., SOCS-1, SOCS-2, CIS, and SOCS-3 react, and when they are overexpressed, they interact with the JAK/STAT signaling pathway. Accordingly, the role of the SOCS-3 protein in the regulation of GH signaling has received increasing attention, and data from these studies has confirmed the close relationship between GH and SOCS-3 at the level of the ovine pituitary. Additional experiments have demonstrated the influence of GH on SOCS-3 mRNA levels. Soon after the discovery of proteins from the SOCS family, Adams et al. [13] showed that SOCS-3 mRNA can be induced by GH in the fibroblast cell line, F442. Moreover, in hepatocytes taken from mice, GH also quickly induced SOCS-3 expression [10]. A significant increase in the expression of SOCS-3 mRNA was also observed in the liver, lung, and muscle of rats without pituitaries that were treated with exogenous GH [4,32]. Due to the overexpression of the SOCS-3 protein strongly inhibited GH signaling [10], it was suggested that increased expression of this regulator could lead to GH resistance during chronic disease and inflammatory processes. Therefore, GH-induced increases in SOCS-3 mRNA expression in the adenohypophyses of sheep in the present experiments may produce similar results.

GH quickly changed the SOCS-3 mRNA levels. We found that incubating ovine pituitary explants with GH modulated the SOCS-3 expression level, which could be observed just 30 min after administration; the maximum effect was observed after 120 min of incubation. These results are consistent with those found in a study in which GH-induced SOCS-3 expression was found to be rapid and transient [10].

The above observations indicate that GH has a stimulatory effect on SOCS-3 expression. However, the results from the present study show that the direction of the GH effects are not as clear-cut as previously concluded. Interestingly, in the tissue samples harvested under LD conditions, we obtained results that showed a pronounced inhibitory impact of GH on the mRNA levels of SOCS-3. These results maybe related to studies on the role of PRL in the expression of SOCS-3 in the pituitary gland of sheep [18]. In these experiments it was found that, in addition to its transcription-stimulating action, PRL may also decrease the levels of SOCS-3 mRNA. This result was found in pituitaries collected in July; the period correlated with the highest concentration of endogenous PRL during the annual cycle in sheep [34]. Low expression levels of the SOCS-3 protein during this period were required to intensify the effects of PRL [18]. The inhibition of SOCS-3 expression under the influence of GH, which was associated with the photoperiod, was also demonstrated in studies with Siberian hamsters that examined the effects of leptin. Leptin significantly reduced the expression levels of hypothalamic SOCS-3 during the SD period compared with the LD period [17]. These studies also demonstrated that endogenous SOCS-3 expression levels in the hypothalamus are higher during the LD season and lower during the SD season [14].

Rousseau et al. [35] demonstrated the importance of a photoperiod for regulating the sensitivity of the hypothalamus to leptin in seasonal species, observing that a reduction in the level of SOCS-3 mRNA during the SD season was associated with a resensitization to leptin signaling. The relationship among season, sensitivity to leptin, and the level of SOCS-3 expression in the Siberian hamster hypothalamus was also studied by Klingenspor et al. [36] and Mercer et al. [37]. The ability of GH to stimulate and inhibit SOCS-3 transcription, demonstrated in the present study, is likely one method by which the biological activity of this pleiotropic ligand is regulated. GH may alter the expression of the SOCS-3 protein in a manner dependent on many factors and may regulate tissue excitability in response to itself.

According to the information presented above, the SOCS protein may be responsible for facilitating the adaptation of the body to environmental conditions to maintain energy homeostasis. Hormone-induced SOCS-3 expression could possibly modulate the relationship among hormones that act through the JAK/STAT signaling pathway. The influence of a given cytokine on SOCS-3 mRNA levels may be similarly related to the existence of many factors, and the metabolic status of the organism may play an important role in modulating these relationships.

## 5. Conclusions

Taken together, our data do not support the hypothesis that leptin modulates GH concentrations in ad libitum-fed sheep.Both leptin and GH influenced the expression of SOCS-3 differently in vitro depending on the dose of ligand and the photoperiod. Together, our studies demonstrated the dominant role of photoperiod on the effect of leptin and GH on the expression of SOCS-3, one of the inhibitors of their signaling pathway.

## Figures and Tables

**Figure 1 animals-12-00403-f001:**
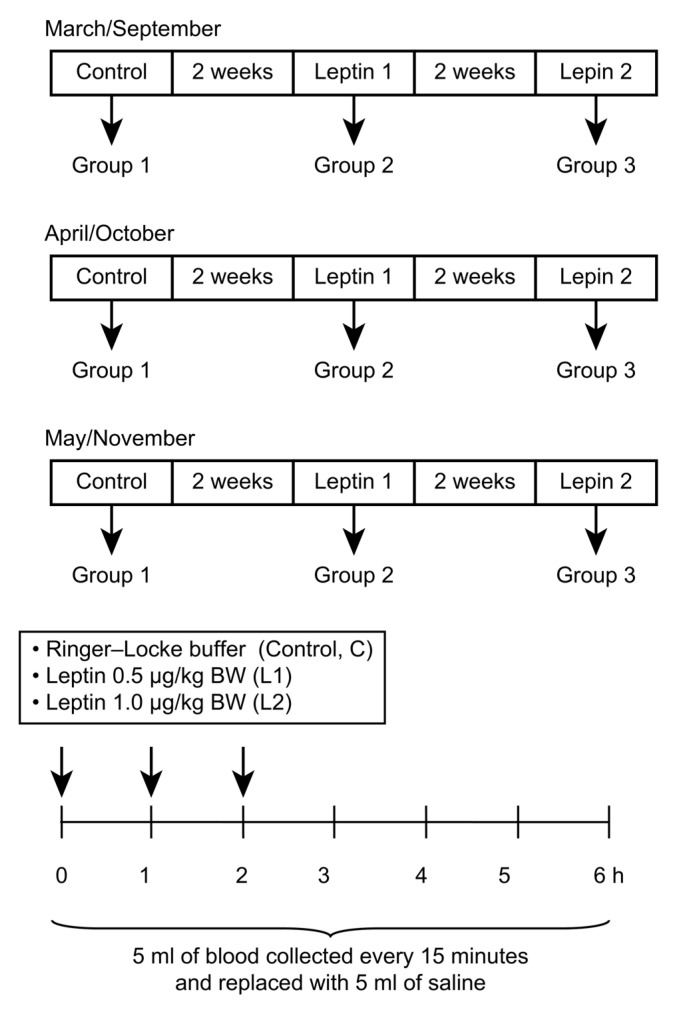
Timeline of the experimental procedure. Twelve ewes were randomly assigned to one of three groups (*n* = 4/group) under long-day (LD; March, April, and May) or short-day (SD; September, October, and November) photoperiods. Treatments consisted of (1) Control, a Ringer–Locke buffer; (2) Leptin 1, a low dose of exogenous leptin (0.5 mg/kg BW); and (3) Leptin 2, a high dose of exogenous leptin (1.0 mg/kg BW), in a switchback design such that four ewes from each group received one of the treatments (Control, Leptin 1, or Leptin 2) centrally in a random order ~2 weeks apart. Arrows indicate central treatments with leptin or Ringer–Locke buffer at 0, 1, and 2 h of the 6-h experiment.

**Figure 2 animals-12-00403-f002:**
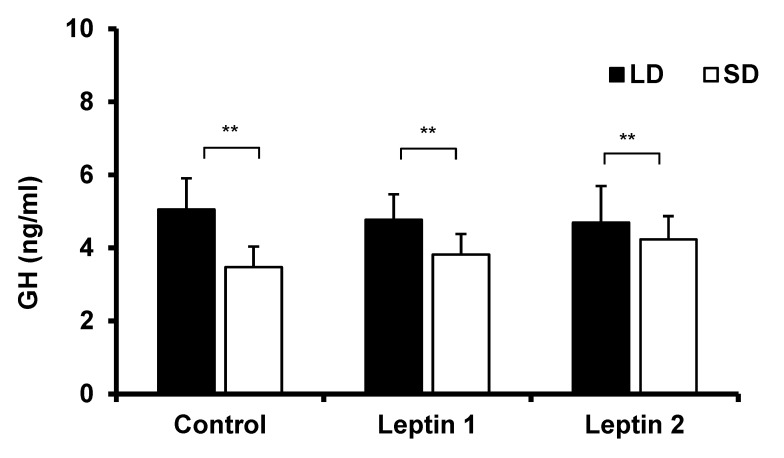
Concentrations of GH in the plasma of IIIV-cannulated ewes treated with Ringer–Locke buffer (Control), 0.5 mg/kg BW leptin (Leptin 1) or 1.0 mg/kg BW leptin (Leptin 2) in both LD and SD photoperiods. Data are presented as the mean(± S.E.M.). Differences are shown with ** (*p* < 0.01). IIIV-third ventricle of brain, LD—long day; SD—short day.

**Figure 3 animals-12-00403-f003:**
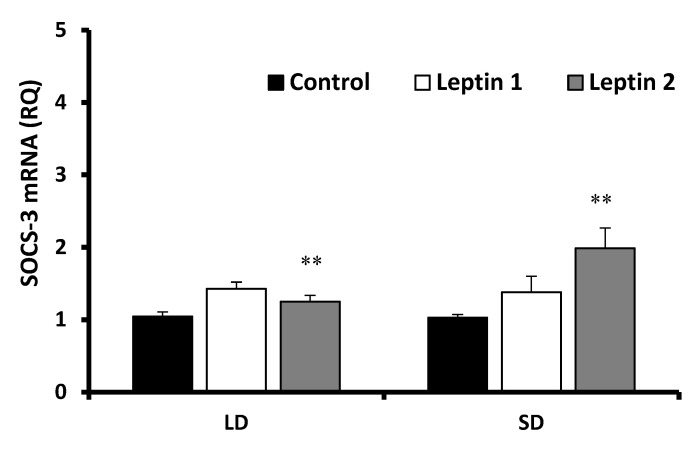
Mean abundance of SOCS-3 transcripts(± SE) in adenohypophysis explants collected during LD and SD seasons affected by leptin at a concentration of 50 ng/mL (Leptin 1) or 100 ng/mL (Leptin 2). SOCS-3 mRNA expression is shown in arbitrary units versus mRNA expression of cyclophilin. The control group was used to calibrate the mean values. ** *p* < 0.01 compared with the Control group. LD—long day, SD—short day.

**Figure 4 animals-12-00403-f004:**
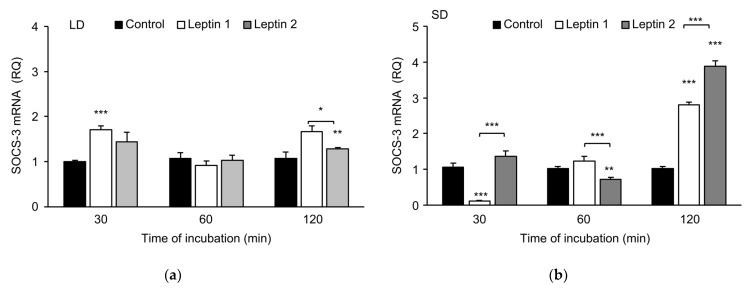
The mean abundance of SOCS-3 transcripts (± SE) in adenohypophysis explants collected during LD (panel A) and SD (panel B) seasons is affected by leptin at concentrations of 50 ng/mL (Leptin 1) or 100 ng/mL (Leptin 2). Panel (**a**) includes data from thelong-day (LD) season; panel (**b**) includes data from the short-day (SD) season. SOCS-3 mRNA expression is shown in arbitrary units versus mRNA expression of cyclophilin. The mean value of the nontreated group (Control) was used to calibrate values. * *p* < 0.05, ** *p* < 0.01 and *** *p* < 0.001 indicate differences versus the Control or among the selected groups. LD—long day; SD—short day.

**Figure 5 animals-12-00403-f005:**
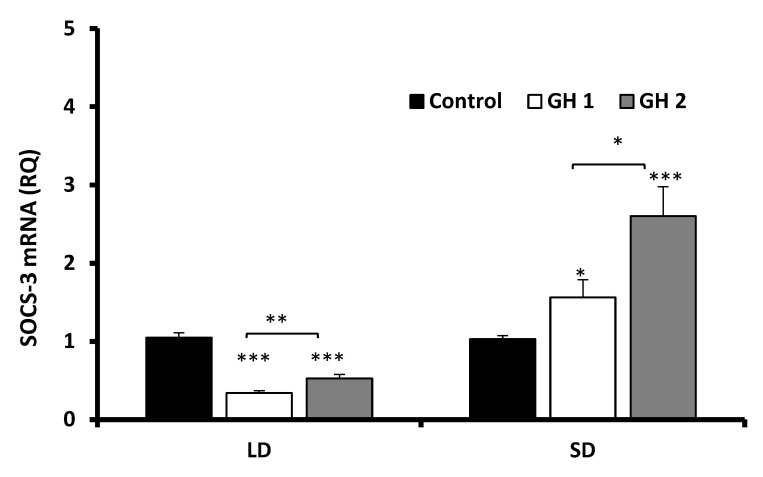
Mean abundance of SOCS-3 transcripts(±SE) in adenohypophysis explants collected during LD and SD seasons incubated with varying concentrations of GH (100 ng/mL (GH 1) or 300 ng/mL (GH 2)). SOCS-3 mRNA expression is shown in arbitrary units versus mRNA expression of cyclophilin. The mean value of the nontreated group (Control) was applied for calibration. * *p* < 0.05, ** *p* < 0.01 and *** *p* < 0.001 indicate differences versus the control group or among the selected groups.

**Figure 6 animals-12-00403-f006:**
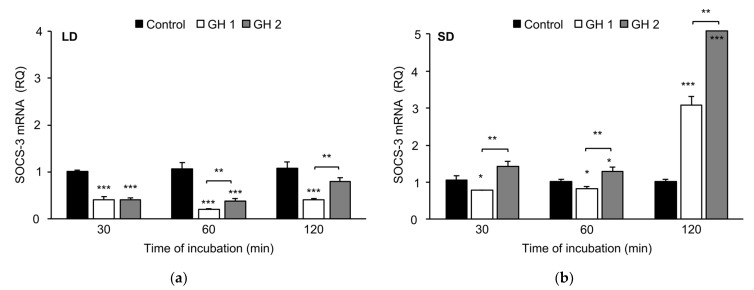
Mean abundance of SOCS-3 transcripts(±SE) in adenohypophysis explants collected during LD (panel **a**) and SD (panel **b**) seasons incubated with varying concentrations of GH (100 ng/mL (GH 1) or 300 ng/mL (GH 2)). SOCS-3 mRNA expression is shown in arbitrary units versus mRNA expression of cyclophilin. The mean value for the nontreated group (Control) was applied for calibration. * *p* < 0.05, ** *p* < 0.01 and *** *p* < 0.001 indicate differences versus the control group or among the selected groups. LD—long day; SD—short day.

**Table 1 animals-12-00403-t001:** Characteristics of primers and probes used to determine cyclophilin (CPH; reference gene) and suppressor of cytokine signaling-3 (SOCS-3; target gene) mRNA expression in sheep.

Gene	Primer Sequence (5′-3′)	Probe Sequence (5′-3′)	Amplicon Size	GenBank Accession Number
CPH	CGGCTCCCAGTTCTTCATCA	FAM-CGTTCCGACTCCGC-MGB	64 bp	D14074
	ACTACGTGCTTCCCATCCAAA			
SOCS-3	CCTCAAGACCTTCAGCTCCAA	FAM-AGCGAGTACCAGCTGG-MGB	68 bp	NM_174466
	CTTGCGCACTGCGTTCAC			

## Data Availability

Data are available from the corresponding author under reasonable request.

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
