# Peer review of "Effects of Leptin, Growth Hormone and Photoperiod on Pituitary SOCS-3 Expression in Sheep"

_animals, 2022, doi:10.3390/ani12030403_

Round 1

Reviewer 1 Report

TITLE: Seasonal Interactions between Leptin and GH and Effects on Pituitary SOCS-3 Gene Expression in Sheep. This title is imprecise. It is unclear whether the independent effects of leptin and growth hormone or only their seasonally-dependent interactions are investigated in this study.

SIMPLE SUMMARY: Similar to the remaining sections of this manuscript, the simple summary would benefit from a round of linguistic revisions by a native speaker or professional editing services. The examples of flawed style and grammar include, but are not limited to, the following phrases and statements: i. The paper investigated relationships... - a research team investigated the relationships, not the paper; ii. The finding that these compounds were significantly influenced by photoperiod... - what exactly was influenced by the photoperiod? Chemical structure of the hormones, their intermediate metabolism or their measurable concentrations?

ABSTRACT: Essentially, the same comment as above could be repeated for this section. For example, the sentence these studies aimed to determine how leptin affects growth hormone (GH) release and 18 investigate the effects of leptin, GH and day length on suppressor of cytokine signaling-3 (SOCS-3) 19 mRNA levels in adenohypophyses of sheep, is not in agreement with the title of the manuscript. Perhaps listing the specific study objectives in the bullet form would help the authors improve this part of their paper.

INTRODUCTION:

L35-38 Superfluous. Delete.

L39 Leptin is produced by white adipose tissue, brown adipose tissue, placenta (syncytiotrophoblasts), ovaries, skeletal muscles, stomach glands, mammary epithelial cells, and bone marrow.

L41-42 ...including the regulation of reproductive and metabolic processes (style).

L43-46 Poor style and inadequate use of technical language (e.g., is regulated by photoperiod instead of depends on the length of the day).

L47-48 If the available information is insufficient and contradictory then there is hardly convincing evidence.

L51-52 Replace the expression concentrations with simply the expression?

L53 Control pathway of what regulatory mechanism?

L57-65 Detailed information on the nature of second messenger systems associated with the activation of GH and leptin receptors is not necessary.

L66 Superfluous. Multiple factors are typically involved in the biological control of hormone action.

L66-80 Revise spelling and punctuation. Delete information pertaining to the mode of GH action.

L84 Grammar (have stated).

L95-96 Style and grammar.

In general, the Introduction does not provide rationale for the study objectives.

MATERIALS AND METHODS:

L107-108 Grammar. The ethics committee did not agree to perform the experimental procedures but evaluated and approved them.

L110-111 Describe the seasonal pattern of reproduction in the Polish Lowland sheep.

L122-136 The authors emphasize the influence of photoperiod on the effects of adipokines and on reproductive cyclicity of ewes, yet by using the ovariectomized estradiol-implanted animals they completely eliminate the influence of ovarian activity on the outcome of their experiments. To minimize the effects of varying estrogen and progesterone levels, the experimental procedures could be conducted in estrus-induced/synchronized females.

L144 The number of animals allotted to different groups is very low. Have the authors confirmed that such numbers were adequate to demonstrate statistically significant differences?

L194 Describe the method of euthanasia.

L245 How were the seasonal differences (day length) incorporated into the statistical model? What were the main effects determined and their interactions?

RESULTS:

The entire section must be thoroughly revised after detailing the specific statistical methods used by the authors.

DISCUSSION:

L353-360 Superfluous.

L360-369 How is the nutritional status of animals relevant to the present study provided that it was restricted to normally fed animals?

L408-428 Not directly relevant to the results of the present study and covered earlier, in part, in the Introduction.

L446-462 Not directly related to the present experiments.

The Discussion and CONCLUSIONS are incoherent and excessively long.

Reviewer 2 Report

  1. There were 7 paragraphs in the part of introduction. However, most of the contents were not really correlated with the research directions in this article. The author didn’t show a good logic for the backgrounds, scientific question, and hypothesis, which should be rewritten completely. Usually, 3-4 paragraphs are enough for the part of introduction.
  2. The experiment design was not described clearly in spite of showing the timeline of the experimental procedures. What’s the meaning for “in a switchback design such that four ewes from each group received one of the treatments (Control, Leptin 1, or Leptin 2) centrally in a random order ~2 weeks apart”? So, how many replicates for each group finally?
  3. In Figure 2, what’s the meaning for “Differences are shown with A, B, and C (P <0.01); a, b, and c (P<0.05); or ** (P <0.01)”? Who compared to who? The symbols of the significances were not labeled clearly or described clearly.

Did A, B, and C show the significant differences among Control, Leptin 1, and leptin 2 groups in the long days?

Did a, b, and c show the significant differences among Control, Leptin 1, and leptin 2 groups in the short days?

As shown in the graph, I don’t think there are significant differences among the 3 groups in the long days or in the short days. The authors should check their statistical analysis carefully.

If so, it is meaningless for the studies in this article.

  1. The author didn’t clarify the relationship among leptin, growth hormone, and SOCS3 expression clearly, so they didn’t have a clear conclusion for their data in this article.
  2. The part of conclusion was written as a background but not a summary.

Round 2

Reviewer 2 Report

The authors have made a great improvement on this article. Even though it is not perfect, I think it can be published on this journal now.